# Image Similarity Metrics Suitable for Infrared Video Stabilization during Active Wildfire Monitoring: A Comparative Analysis

**Mario M. Valero** [1], **Steven Verstockt** [2], **Christian Mata** [1], **Dan Jimenez** [3], **Lloyd Queen** [4], **Oriol Rios** [1], **Elsa Pastor** [1] **and Eulàlia Planas** [1,*]

1   Centre for Technological Risk Studies, Universitat Politècnica de Catalunya, 08034 Barcelona, Spain; mm.valero@pm.me (M.M.V.); christian.mata@upc.edu (C.M.); oriol.rios@upc.edu (O.R.); elsa.pastor@upc.edu (E.P.)
2   IDLab, Ghent University – imec, 9502 Ghent, Belgium; steven.verstockt@ugent.be
3   Missoula Fire Sciences Lab, US Forest Service Rocky Mountain Research Station, Missoula, MT 59808, USA; dan.jimenez@usda.gov
4   National Center for Landscape Fire Analysis, University of Montana, Missoula, MT 59812, USA; lloyd.queen@mso.umt.edu
*   Correspondence: eulalia.planas@upc.edu

**Abstract:** Aerial Thermal Infrared (TIR) imagery has demonstrated tremendous potential to monitor active forest fires and acquire detailed information about fire behavior. However, aerial video is usually unstable and requires inter-frame registration before further processing. Measurement of image misalignment is an essential operation for video stabilization. Misalignment can usually be estimated through image similarity, although image similarity metrics are also sensitive to other factors such as changes in the scene and lighting conditions. Therefore, this article presents a thorough analysis of image similarity measurement techniques useful for inter-frame registration in wildfire thermal video. Image similarity metrics most commonly and successfully employed in other fields were surveyed, adapted, benchmarked and compared. We investigated their response to different camera movement components as well as recording frequency and natural variations in fire, background and ambient conditions. The study was conducted in real video from six fire experimental scenarios, ranging from laboratory tests to large-scale controlled burns. Both Global and Local Sensitivity Analyses (GSA and LSA, respectively) were performed using state-of-the-art techniques. Based on the obtained results, two different similarity metrics are proposed to satisfy two different needs. A normalized version of Mutual Information is recommended as cost function during registration, whereas 2D correlation performed the best as quality control metric after registration. These results provide a sound basis for image alignment measurement and open the door to further developments in image registration, motion estimation and video stabilization for aerial monitoring of active wildland fires.

**Keywords:** wildland fire; remote sensing; infrared imagery; video stabilization; image registration; sensitivity analysis; image similarity

## 1. Introduction

Forest fires have been studied through remote sensing techniques for decades. A number of spaceborne sensors have successfully been used to analyze various fire aspects and post-fire effects [1]. Existing applications include the detection of active fires [2–4], burned area measurement [5–9], sensing of radiated energy [10,11] and the estimation of pyrogenic gas emissions [12,13], among others.

Similarly, airborne imaging systems are being increasingly employed to gain detailed insight into fire behavior variables such as fire rate of spread, fire line intensity and fire radiative power [14–22]. Unmanned and remotely piloted aircraft further simplify sensor deployment while significantly reducing operation costs and risk [23–25].

Although a few successful experiences have been reported that use airborne monitoring systems in large-scale wildfires [17,26], the majority of developments in fire detection and monitoring occur via sensing prescribed fires, which are often restricted in areal extent as well as fire line radiative intensity [27]. In these cases, the remote sensor is usually placed in a fixed position or a hovering aircraft and it is deployed to collect high spatial resolution images with a moderate temporal resolution for the full duration of flaming combustion [28–30]. This type of deployment rarely follows a large-area mapping mission profile where parallel and overlapping flight lines are required. Nevertheless, turbulence from the fire often results in significant roll, pitch and yaw variations that are hard to cancel with mechanical stabilization systems only. Given camera motion during the acquisition, image registration and rectification are required before spatial inference can be completed (for example, to measure fire residence time per pixel or rate of spread).

Within sensor types suitable for wildfire monitoring, optical cameras working in the thermal infrared (TIR) range are widely applied to characterize active fire behavior due to their high availability and versatility [11,15,17,31,32]. Airborne TIR cameras allow measuring fire geometry and radiated energy with high spatial and moderate temporal resolution even in the presence of smoke. Due to these advantages, several TIR image processing algorithms have been developed for automated computation of fire behavior metrics [15,20,21]. Automated methodologies allow not only faster but also more rigorous quantitative studies by removing bias and ensuring a systematic analysis framework. In order to draw meaningful conclusions, fire behavior metrics must be measured explicitly in time and space, avoiding long-term and wide-field average values whenever possible.

However, a number of limitations remain in the automated processing of fire thermal infrared imagery. Among existing needs, image registration tools that allow camera motion estimation and cancellation are in high demand [24,33]. The current approach used to georeference aerial TIR fire imagery is based on the manual annotation of ground control points in every video frame [15,18,21,34]. This methodology is not only very time consuming but also prone to errors and hard to implement operationally. Important difficulties with image georeferencing have been reported in previous studies, sometimes resulting in loss of data [18,24]. Because of the highly variable energy emitted by the fire and the dynamic radiance range used by many cameras, background objects are sometimes not well resolved. This fact prevents the identification of GPSed ground control points in the video. Even when it is successfully performed, manual georeferencing has been identified as one of the most significant sources of uncertainty in the study of wildfire behavior from aerial TIR imagery [22]. These limitations seriously restrict the amount of quantitative information obtainable through remote sensing as well as its quality. Consequently, the achievement of accurate automated image georeferencing is a high priority necessity for wildfire science.

A fundamental operation during image registration is the measurement of image similarity, usually with the ultimate goal of maximizing such measure. There are three major types of dissimilarities that can be observed when comparing two or more images [35]. The first type is misalignment, which appears due to variations in the position of the acquisition sensor. These variations are relatively easy to model as geometric transformations. Frequently, prior knowledge of the scene determines the class of transformations to be explored, and this selection in turn determines the most suitable registration method. The second type of image dissimilarities are generated by variations in external conditions during image acquisition. Contrary to first-type image differences, second-type dissimilarities are frequently not easy to model. Lighting and atmospheric conditions, among others, fall within this category. Finally, the third type of image differences are caused by changes in the scene itself. The observer is usually interested in these changes, which include movement of the objects under study, among others.

The main objective of image registration techniques is to find the correct spatial transformation that cancels the first type of image dissimilarities, without being affected by variations of the second type. By doing this, the third type of image differences are then easier to analyze. Differences of types two and three are not cancelled by registration methods but constitute a challenge for them because they prevent an exact match between images that must be compared. In the wildfire context, the movement of a drone operating the camera constitutes an example of type-one variations. Conversely, differences in lighting conditions and smoke concentration between camera and fire fall within the second category because they produce variations in the imaging scenario and they can affect image processing algorithms. Finally, changes in the fire itself, which is dynamic, are the best example of type-three image dissimilarities.

In contrast with visible imagery, fire thermal infrared video entails several challenges that have so far prevented the achievement of automated image registration. Fire monitoring requires high measurement ranges for brightness temperature, usually starting over 200 °C. This fact diminishes the amount of detail distinguishable in the cold background. Moreover, fire usually occupies a large portion of the camera field of view. Because fire is dynamic, this fact significantly hinders the identification of persistent features between images acquired at different times.

This article analyzes the problem of image similarity measurement in the context of forest fire aerial remote sensing, specifically focusing on TIR imagery of active fires acquired from a vantage point and an oblique perspective. The ultimate goal of this study is the identification of image similarity metrics suitable for inter-frame registration and video stabilization. State-of-the-art methodologies used to measure image similarity in other fields were surveyed and benchmarked. Tested methods include metrics based on gray value difference, gray value correlation and information theory. Metrics were assessed based on their ability to meet two specific needs: on the one hand, a well-behaving cost function is needed during registration; on the other, a robust estimator of absolute image alignment is required after registration for quality control.

## 2. Background: Image Similarity Metrics

The most popular approach to measure image similarity has historically been based on gray difference statistical metrics such as intensity mean squared difference and two-dimensional correlation [35–37]. Cross-correlation has been used for image registration during decades and it is still in use in several applications, including remote sensing [38–40]. Recently the use of direct gray difference measurements has decayed in favor of more powerful metrics based on information theory, such as Mutual Information (MI) [39,41,42]. Translating gray level values into the more general measure of information content provides enormous flexibility. For this reason, MI is vastly employed not only for image similarity measurement but also to fuse multispectral [39,43,44] and multi-modal [45–47] information. However, this flexibility entails an elevated computational cost that may become prohibitive under certain circumstances. Speed requirements (e.g., for real-time processing) and hardware limitations (e.g., for deployment aboard satellites or unmanned aircraft) usually motivate the use of low-complexity algorithms [48–50]. In addition to this wide variety of advantages and drawbacks, the performance of each methodology varies significantly with the field of application. Fire IR imagery presents important singularities with respect to other remote sensing and computer vision scenarios. In order to address this issue, we analyzed the suitability for fire monitoring of some of the most widely employed image similarity metrics.

### 2.1. Intensity 2D Correlation

Cross-correlation and the two-dimensional correlation coefficient are statistical indices widely used in image registration [35]. The 2D correlation coefficient between two images $corr2D(I_1, I_2)$ (Equation (1)) provides a scalar measurement of their global similarity, whereas cross-correlation $C(u, v)$ measures the degree of similarity between a reference image $I$ and a template $T$ shifted $u$ and

$v$ pixels in the $x$ and $y$ direction, respectively (Equation (2)). Cross-correlation is frequently used for template matching and pattern recognition.

$$corr2D(I_1, I_2) = \frac{covariance(I_1, I_2)}{\sigma_1 \sigma_2} = \frac{\sum_i \sum_j (I_1(i,j) - \mu_1)(I_2(i,j) - \mu_2)}{\sqrt{\sum_i \sum_j (I_1(i,j) - \mu_1)^2)(I_2(i,j) - \mu_2)^2)}} \tag{1}$$

$$C(u,v) = \frac{\sum_x \sum_y T(x,y) I(x-u, y-v)}{\sqrt{\sum_x \sum_y I^2(x-u, y-v)}} \tag{2}$$

In Equation (1), $\mu_i$ and $\sigma_i$ represent, respectively, the gray value average and standard deviation within each image.

The 2D correlation coefficient has two important advantages. First, it provides a similarity measurement in the fixed range $[-1, 1]$. Secondly, it shows a linear relationship with image similarity under certain statistical assumptions [35]. Both properties are particularly useful in an image registration scheme because they allow an absolute assessment of the achieved registration quality. This way, the estimated registration transformation can be accompanied by a confidence assessment. A correlation coefficient of 1 represents a perfect match, achieved when identical images are perfectly aligned.

### 2.2. Intensity Mean Squared Difference (IMSD)

Intensity mean squared difference (Equation (3)) falls within the group of the simplest metrics useful to measure dissimilarity between two variables. It is simple and computationally efficient, and it also provides an absolute similarity estimation. In this case, perfect match is denoted by an IMSD value of 0.

$$IMSD(I_1, I_2) = \frac{1}{N \cdot M} \sum_{i=1}^{N} \sum_{j=1}^{M} (I_1(i,j) - I_2(i,j))^2 \tag{3}$$

In Equation (3), $N$ and $M$ indicate the number of rows and columns in images $I_1$ and $I_2$, which must both be the same size.

### 2.3. Mutual Information

Mutual Information (MI) quantifies the dependence between two variables, more specifically the amount of information that one variable contains about the other [51]. This definition allows for a criterion frequently used in image registration problems, which states that two images are geometrically aligned when MI between the intensity values of corresponding pixels -or voxels- is maximal [52].

Image similarity metrics based on Mutual Information were first proposed by Viola [53] and Collignon et al. [54]. Since then, they have been extensively used in the field of medical imaging [41,46,52,55] and, more recently, in remote sensing [39,42,44,56].

If $A$ and $B$ are two random variables with marginal probability distributions $p_A(a)$ and $p_B(b)$, the Mutual Information between them $I(A, B)$ measures the distance between their joint distribution $p_{AB}(a,b)$ and the joint distribution they would have if they were completely independent, $p_A(a) \cdot p_B(b)$. This distance represents the degree of dependence between $A$ and $B$ and it is usually computed using the Kullback–Leibler measure (Equation (4)).

$$I(A, B) = \sum_{a,b} p_{AB}(a,b) \, log \frac{p_{AB}(a,b)}{p_A(a) \cdot p_B(b)} \tag{4}$$

Alternatively, MI can be defined using the concept of image entropy. Entropy ($H$) measures the uncertainty of a random variable. It is widely used in information theory and its most common mathematical definition was proposed by Shannon [57] (Equation (5)).

$$H = -\sum_i p_i \, log \, p_i \tag{5}$$

It can be demonstrated [41] that Equation (4) can be rewritten as Equations (6)–(8) introducing Shannon entropy:

$$I(A, B) = H(A) + H(B) - H(A, B) \tag{6}$$
$$= H(A) - H(A \mid B) \tag{7}$$
$$= H(B) - H(B \mid A) \tag{8}$$

where $H(A)$ and $H(B)$ are the entropy of images $A$ and $B$, respectively, $H(A, B)$ is their joint entropy, $H(A \mid B)$ is the conditional entropy of $A$ given $B$ and $H(B \mid A)$ is the conditional entropy of $B$ given $A$. $H(A)$ measures the uncertainty of $A$, whereas $H(A \mid B)$ represents the amount of uncertainty left in $A$ when knowing $B$. Consequently, $I(A, B)$ can be understood as the reduction in uncertainty of $A$ caused by the knowledge of $B$. In other words, $I(A, B)$ represents the amount of information that $B$ contains about $A$.

Mutual Information is very powerful because it allows a general comparison of two images without assumptions about their nature or the nature of their relation, and with no need for prior segmentation. This provides the additional capability of comparing images of a different nature, a property that has been exploited for image fusion purposes [58,59]. Furthermore, MI presents some significant advantages over metrics based on cross-correlation, which are affected by changes in lighting conditions and reflectance dependence on wavelength [39,42].

Despite these good properties, there are also some drawbacks related to Mutual Information. The high computational cost of MI computation, together with interpolation artefacts and the relatively high amount of noise present in the MI surface and its derivatives when it is undersampled [39], hinder convergence of MI-based registration algorithms. Additionally, the MI registration function may contain local maxima, which can result in misregistration [60,61].

Furthermore, the original MI formulation presents two significant limitations when used as image similarity metric alone, decoupled from the image registration framework. On the one hand, MI value is sensitive to the amount of overlap between compared images [41,62]. On the other hand, it does not provide an absolute measurement of how well two images are aligned. MI can estimate relative agreement between two images: the more similar two images are, the greater their MI value. However, this MI value cannot be directly compared against an absolute similarity scale and identical images do not always reach the same MI value. The main consequence of this is that the quality of a certain registration algorithm cannot be absolutely measured using MI, and therefore there is no means of verifying whether the achieved optimum MI is acceptable.

## 2.4. Normalized Mutual Information (NMI)

Several improvements have been proposed to overcome MI limitations. Studholme et al. [62] suggested a revised version of MI, invariant to image overlap (Equation (9)). Although the original authors called this revision *normalized*, its value is in fact not comprised in the interval $[0, 1]$ [59]. Therefore, we refer to it as Studholme's Mutual Information (SMI).

$$\text{SMI}(A, B) = \frac{H(A) + H(B)}{H(A, B)} \tag{9}$$

Similar to SMI is the so-called *entropy correlation coefficient* (ECC, Equation (10)), first proposed by Astola and Virtanen [63] and tested by Maes et al. [52]. The behavior of ECC is similar to SMI by

definition [41,61] and Maes et al. [52] did not find a clear difference in performance when compared with original MI.

$$\text{ECC}(A, B) = \frac{2\, I(A, B)}{H(A) + H(B)} \tag{10}$$

In this paper, we propose an alternative MI formulation that provides actual normalization and, consequently, an absolute measurement of image similarity. We assessed the performance of the Normalized Mutual Information (NMI) as defined in Equation (11). This formulation had previously been used in machine learning algorithms [64] but it had barely received attention in image analysis problems, being Bai et al. [65] and Pillai and Vatsavai [66] the only two exceptions we found. This formulation is based on the fact that $H(X) = I(X, X)$ and constitutes an analogy with a normalized inner product in Hilbert space. NMI is symmetric and restricted to the range [0,1] by definition.

$$\text{NMI}(A, B) = \frac{I(A, B)}{\sqrt{H(A)H(B)}} \tag{11}$$

Estévez et al. [67] took a similar approach when comparing a feature set *F* with a subset of itself *S*. They defined the normalized mutual information between $f_i \in F$ and $f_s \in S$ by dividing their MI by the minimum entropy of both sets (Equation (12)):

$$\text{NMI}_{\text{Estevez}}(f_i; f_s) = \frac{I(f_i; f_s)}{min\,\{H(f_i), H(f_s)\}} \tag{12}$$

$\text{NMI}_{\text{Estevez}}$ is also symmetric and takes values in [0,1]. However, its formulation is less convenient for our image registration problem. $\text{NMI}_{\text{Estevez}}$ similarity values may change abruptly when modifying the reference frame, and consequently the $\text{NMI}_{\text{Estevez}}$ distribution is subject to discontinuities along a video sequence. Updating the reference frame is especially important during wildfire video stabilization to account for fire evolution. Therefore, we suggest using the NMI formulation shown in Equation (11).

## 3. Methodology

Similarity metrics described in Section 2 were subject to Global and Local Sensitivity Analyses (GSA and LSA, respectively) in order to assess their response to different variables of interest such as camera movement, video temporal resolution and natural variations in fire, background and ambient conditions. GSA was conducted first for general screening of significant variable relationships. Subsequently, metric sensitivity was studied locally in the region of interest where similarity must be measured. This local region consists of a limited range of camera translations, rotations and scaling around the nominal recording point of zero perturbation. Details about the GSA and LSA workflows are provided in Sections 3.3 and 3.4, respectively.

These analyses were performed on six TIR video sequences of active spreading fire recorded during laboratory and field experiments. All cameras were installed at fixed elevated vantage points so that the obtained video was stable. Recorded TIR video was used as reference and frames were perturbed through synthetic image translations, rotations and changes in scale that produced virtual camera movement. Individual experiment, camera and footage details are described in Section 3.1. Due to its higher computational cost, GSA was limited to four video sequences representative of different experimental setups, whereas LSA was performed for all available footage. Figure 1 summarizes the followed methodology.

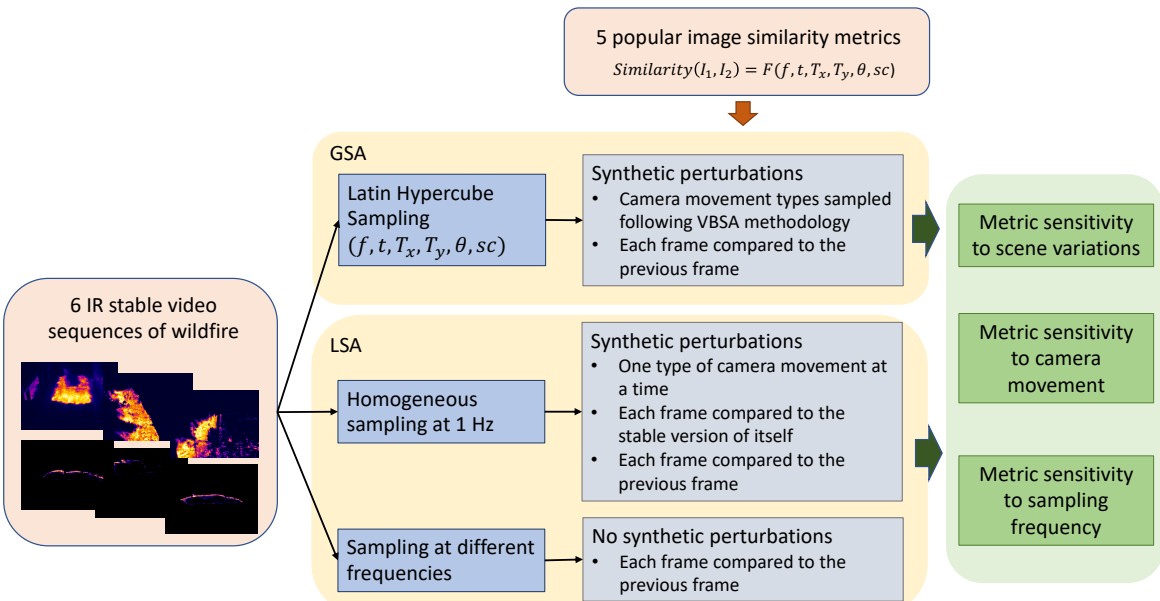

**Figure 1.** Design of the comparative analysis conducted to assess the behavior of various image similarity metrics. Desired properties were high sensitivity to camera movement and low sensitivity to scene variations and sampling frequency. LSA was applied to the 6 available video sequences. GSA was applied to sequences 1, 2, 4 and 5 due to computational restrictions.

### 3.1. Test Data

GSA and LSA studies were applied to six experimental scenarios ranging from small laboratory tests to large-scale field experiments. In all cases, fire evolution was recorded from fixed vantage points using thermal infrared cameras. Employed cameras and setups varied, but all video sequences were stable. The resulting dataset allowed a systematic study under controlled, yet dissimilar, conditions.

*Scenario 1* was recorded in the Centre for Technological Risk Studies at Universitat Politècnica de Catalunya. A homogeneous bed of straw was burned on a 1.5 m × 3 m combustion table to reproduce fire spread on a flat horizontal surface with no wind. *Scenarios 2* and *3* were recorded at the Tall Timbers Research Station in Tallahassee, FL, USA, in April 2017. These video sequences were acquired during a set of small-scale experimental burns on mixed rough/long leaf pine fuels. *Scenarios 4–6* were monitored during one of the most complete large-scale experimental campaigns conducted so far, RxCADRE [29]. Video sequences 4, 5 and 6 correspond to plots S3, S4 and S5 of this experiment, respectively. These three plots were recorded with a high-resolution IR camera mounted on a boom lift [68]. Burned vegetation was a mix of grass and shrubs, predominantly turkey oak. Figure 2 shows sample frames from all scenarios, while Table 1 summarizes the technical details of the employed thermal cameras. Figure 3 provides additional information about the experimental setups.

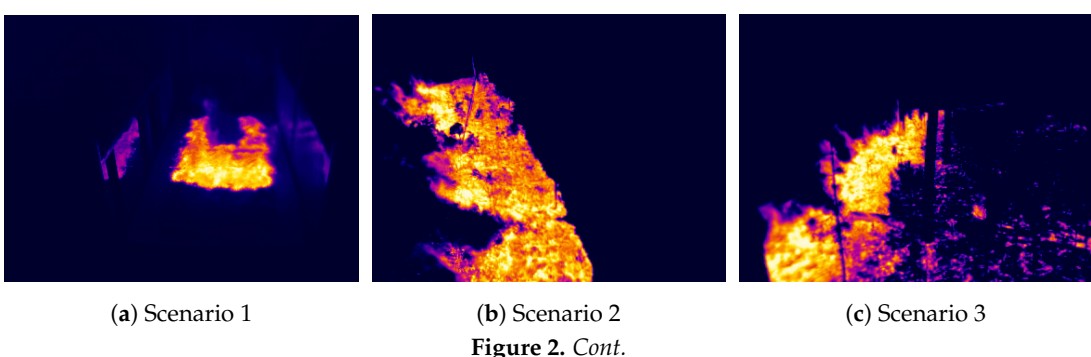

(**a**) Scenario 1　　　　　　　(**b**) Scenario 2　　　　　　　(**c**) Scenario 3

**Figure 2.** *Cont.*

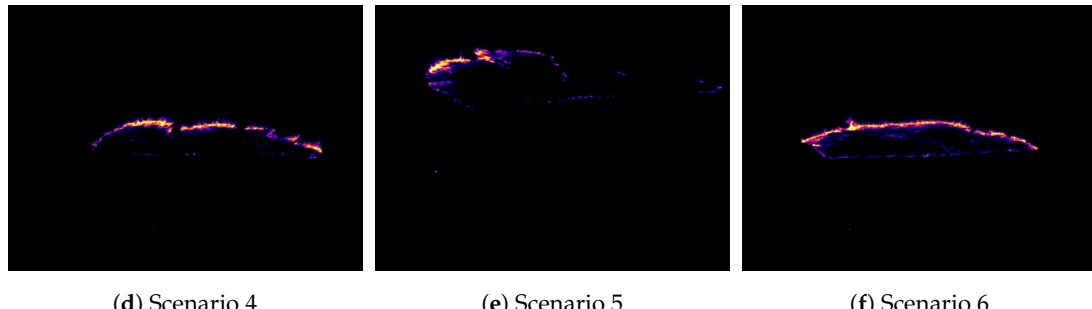

(**d**) Scenario 4          (**e**) Scenario 5          (**f**) Scenario 6

**Figure 2.** Sample frames of the six video sequences (**a**–**f**) employed in this study.

**Table 1.** Camera properties and parameters used to record analyzed footage.

| Scenario | Camera Commercial Name | Spectral Range Wavelength (μm) | Brightness Temperature Range (°C) | Image Resolution (Pixels) | Field of View (°) | Thermal Sensitivity (mK) | Recording Frequency (Hz) |
|---|---|---|---|---|---|---|---|
| 1 | Optris PI 640 | [7.5, 13] | [20, 900] | 640 × 480 | 60 × 45 | 75 | 32 |
| 2 | Optris PI 400 | [7.5, 13] | [200, 1500] | 382 × 288 | 60 × 45 | 75 | 27 |
| 3 | Optris PI 400 | [7.5, 13] | [200, 1500] | 382 × 288 | 60 × 45 | 75 | 27 |
| 4 | FLIR SC660 | [7.5, 13] | [300, 1500] | 640 × 480 | 45 × 30 | 30 | 1 |
| 5 | FLIR SC660 | [7.5, 13] | [300, 1500] | 640 × 480 | 45 × 30 | 30 | 1 |
| 6 | FLIR SC660 | [7.5, 13] | [300, 1500] | 640 × 480 | 45 × 30 | 30 | 1 |

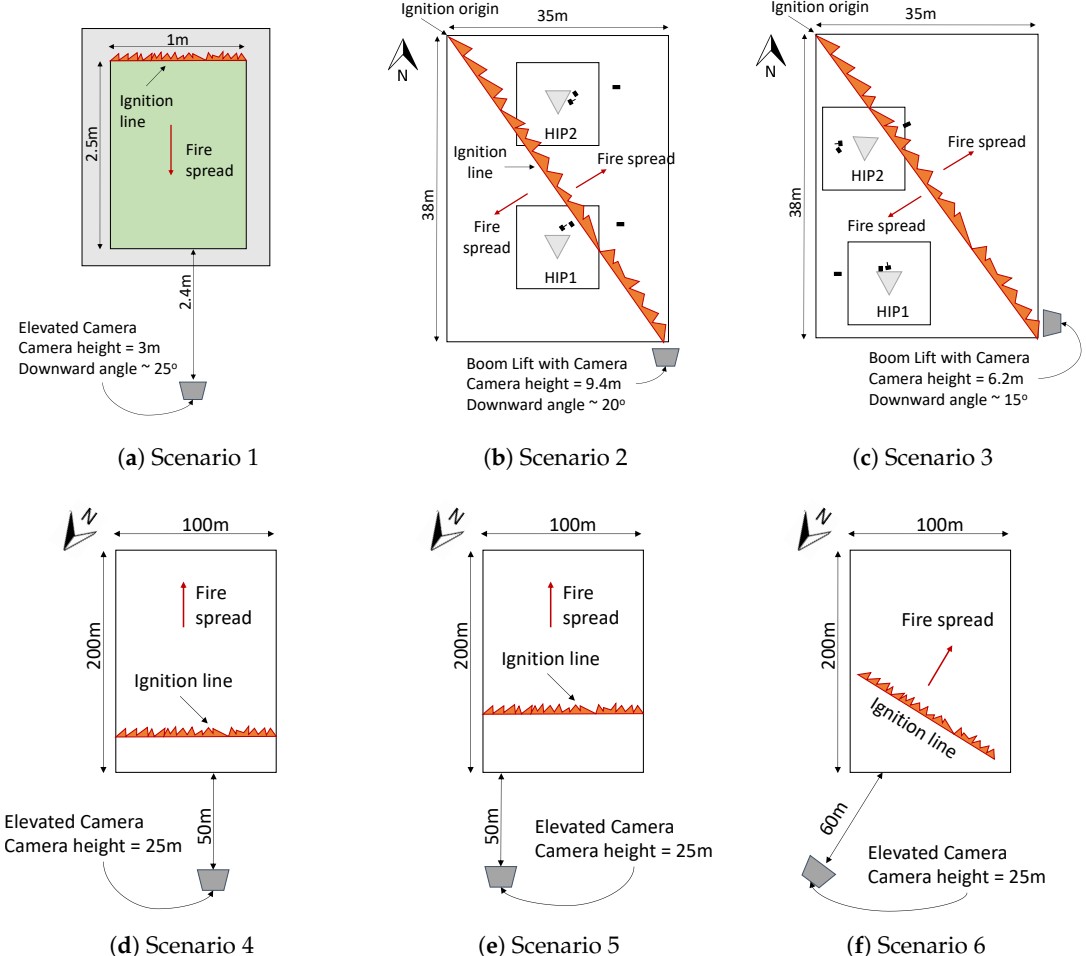

(**a**) Scenario 1          (**b**) Scenario 2          (**c**) Scenario 3

(**d**) Scenario 4          (**e**) Scenario 5          (**f**) Scenario 6

**Figure 3.** Experimental setups used to record the six video sequences (**a**–**f**) employed in this study.

Crown fire did not occur in any of the presented scenarios. Still, this dataset can be considered representative of the typical use of thermal IR cameras in forest fire research.

*3.2. Approach Overview*

Sensitivity analysis is defined as *the study of how uncertainty in the output of a model can be apportioned to different sources of uncertainty in the model input* [69]. In other words, it attempts to quantify the effect that deviations in input parameters have on model outputs. In modeling literature, this has traditionally been achieved in practice through the estimation of partial derivatives of a particular model output versus a particular input. Because such derivatives must be estimated locally at a point of interest, this approach is usually referred to as local sensitivity analysis (LSA).

While LSA may provide valuable insight into model behavior, the scientific discipline of sensitivity analysis has evolved towards more general terms. Statistical studies, risk analysis and reliability assessments—just to name a few—require a broader approach in which the influence of factors on outputs is studied along the entire input space. Consequently, modern global sensitivity analyses are often based on Monte Carlo space sampling and the computation of statistical measures [70].

To apply sensitivity analysis techniques to the study of image similarity measurement, one can understand metrics as models that allow computing a certain output of interest—i.e., similarity—from a set of inputs—i.e., two images. Furthermore, one can disaggregate the actual input parameters into simpler components. In the specific case of active fire video stabilization, the two images to be compared will typically be extracted from the same video sequence by sampling the video at a given frequency. Time elapsed between both frames, which is related to sampling frequency, can significantly affect image similarity, mostly due to fire dynamics. Moreover, we can assume the second image misaligned with respect to the first one if the camera moved between both acquisitions. Such misalignment can be expressed in terms of a two-dimensional relative translation, a rotation angle and a scale coefficient—if only affine transformations are considered. Beyond sampling frequency and misalignment, similarity values computed between the first frame and the second may vary greatly with the state of the scene. Among others, the portion of the field of view that is covered by fire and the flaming intensity can importantly affect similarity metric behavior. In this study, we grouped all factors contributing to absolute changes in the state of the scene under the variable *time*. *time* represents the time at which the first frame was acquired, measured with respect to whichever time reference is selected—typically, the start of the video sequence.

Therefore, the similarity value provided by any of the investigated metrics can be understood as a function of sampling frequency ($f$), time ($t$) and geometric misalignments—relative translations ($T_x, T_y$), rotation ($\theta$) and scaling ($sc$) (Equation (13)). This function is our model, and we studied the sensitivity of its only output—similarity—to each of its inputs, which were assumed independent of each other. The explored input space is summarized in Table 2.

$$\text{Similarity}(I_1, I_2) = F(f, t, T_x, T_y, \theta, sc) \tag{13}$$

**Table 2.** Input parameter ranges considered for sensitivity analysis. Time ranges were set as wide as possible, provided that fire was present in the scene. Maximum sampling frequency corresponds to video recording frequency.

| Video Sequence | Translation Range (% of Width/Height) | Rotation Range (deg) | Scaling Range | Frequency Range (Hz) | Time Range (s) |
|---|---|---|---|---|---|
| 1 | [−20, 20] | [−25, 25] | [0.8, 1.2] | [0.1, 32] | [60, 240] |
| 2 | [−20, 20] | [−25, 25] | [0.8, 1.2] | [0.1, 27] | [8, 660] |
| 3 | [−20, 20] | [−25, 25] | [0.8, 1.2] | [0.1, 27] | [23, 700] |
| 4 | [−20, 20] | [−25, 25] | [0.8, 1.2] | [0.1, 0.86] | [90, 1560] |
| 5 | [−20, 20] | [−25, 25] | [0.8, 1.2] | [0.1, 0.88] | [45, 700] |
| 6 | [−20, 20] | [−25, 25] | [0.8, 1.2] | [0.1, 0.87] | [90, 770] |

### 3.3. Global Sensitivity Analysis

Among existing GSA methods, the most powerful approach consists of estimating the conditional variance of model outputs with respect to each of its inputs. This strategy, named Variance-Based Sensitivity Analysis (VBSA), allows not only understanding model sensitivity but also quantifying it. Saltelli et al. [70] defined conditional variance of a model output $Y$ as its variance when one of the inputs $X_i$ is fixed to a specific value $x_i^*$. This can be represented as $V_{X_{\sim i}}(Y|X_i = x_i^*)$, where $V_{X_{\sim i}}$ indicates that the resulting variance is taken over all factors but $X_i$.

This definition of conditional variance leads to the formulation of two types of sensitivity indices widely used in GSA: first-order and total sensitivity indices. The first-order sensitivity index of $X_i$ on $Y$, $S_i$ (Equation (14)), measures the variance produced in $Y$ when only $X_i$ is modified. This effect is averaged over all possible values of $X_i$ to provide a general measure not limited to a specific point in the input space. The reader is referred to Saltelli et al. [70] for the complete mathematical derivation of this metric.

$$S_i = \frac{V_{X_i}\left(E_{X_{\sim i}}(Y|X_i)\right)}{V(Y)} \tag{14}$$

First-order sensitivity indices do not take into consideration potential input interactions, which may be relevant in non-linear models. In order to account for them, higher-order indices can be defined analogously to Equation (14). In practice, although higher-order indices can be estimated, their detailed computation has an important drawback. The amount of sensitivity indices increases exponentially with the number of inputs. Specifically, a system with $k$ parameters will have $2^k - 1$ indices including first-order and higher-order terms. The computation of all terms is usually impractical, especially because this detailed information can be replaced with an indirect measurement of higher-order effects through the so-called total effects.

The total effect of factor $X_i$ is defined as the sum of all terms of any order that include $X_i$. In other words, $S_{Ti}$ encompasses all possible contributions of $X_i$—both direct and indirect—to the output variance. This can be expressed through Equation (15):

$$S_{Ti} = 1 - \frac{V\left(E\left(Y|X_{\sim X_i}\right)\right)}{V(Y)} = \frac{E\left(V\left(Y|X_{\sim X_i}\right)\right)}{V(Y)} \tag{15}$$

where $\dfrac{V\left(E\left(Y|X_{\sim X_i}\right)\right)}{V(Y)}$ combines all terms of any order that do not include factor $X_i$.

According to Saltelli et al. [70] (also stated in Saltelli et al. [69] and references therein), the computation of all first-order indices and total effects of a model provides sufficient characterization of its sensitivity pattern while keeping computational cost acceptable in most cases.

Still, estimation of variances and expected values requires a high amount of model runs for a meaningful sample of the input space. Such computation may become unfeasible for complex models, which has motivated the development of alternative methods to gain insight into model sensitivity. Example algorithms developed to find approximate sensitivity information include the Elementary Effect Test, Monte Carlo Filtering and the Fourier Amplitude Sensitivity Test [71].

Due to the manageable cost of computing similarity between two images, VBSA could be applied in this study, although it was limited to four TIR sequences, namely scenarios 1, 2, 4 and 5. These 4 scenarios were considered representative of different fire scales and conditions.

Our implementation followed recommendations given by Saltelli et al. [70]. These authors claim to provide the best algorithm available today to compute first-order and total-effect indices purely from model evaluations. Their method builds on the original approach proposed by Sobol [72] and is based on the following steps:

1.  Generate a sample of the model input space of size $2N$. This can be accomplished through random sampling or using sequences of quasi-random numbers. The latter approach allows a significant reduction on the sample size necessary to achieve convergence in estimated statistics.

2.　Split the input sample into two groups. The result will be two matrices of size $N \times M$, where $M$ is the number of model inputs. We call these matrices $A$ and $B$.

3.　Create a third matrix $C$ by combining columns from $A$ and $B$. Specifically, $C$ will be a vertical concatenation of $M$ submatrices $C_i$, where each $C_i$ is composed of all columns of $B$ except the $i$th column, which is taken from $A$.

4.　Run the model for each sample in matrices $A$, $B$ and $C$, thus obtaining output vectors $Y_A$, $Y_B$ and $Y_C$.

5.　Compute first-order ($S_i$) and total-effect ($S_{Ti}$) sensitivity indices defined in Equations (14) and (15). $S_i$ and $S_{Ti}$ can be computed from vectors $Y_A$, $Y_B$ and $Y_C$ using Equations (16) and (17), respectively.

$$S_i = \frac{V\left(E\left(Y|X_i\right)\right)}{V(Y)} = \frac{Y_A \cdot Y_{C_i} - f_0^2}{Y_A \cdot Y_A - f_0^2} = \frac{\frac{1}{N}\sum_{j=1}^{N}\left(y_A^j y_{C_i}^j - f_0^2\right)}{\frac{1}{N}\sum_{j=1}^{N}\left(\left(y_A^j\right)^2 - f_0^2\right)} \tag{16}$$

$$S_{Ti} = 1 - \frac{V\left(E\left(Y|X_{\sim i}\right)\right)}{V(Y)} = 1 - \frac{Y_B \cdot Y_{C_i} - f_0^2}{Y_A \cdot Y_A - f_0^2} = 1 - \frac{\frac{1}{N}\sum_{j=1}^{N}\left(y_B^j y_{C_i}^j - f_0^2\right)}{\frac{1}{N}\sum_{j=1}^{N}\left(\left(y_A^j\right)^2 - f_0^2\right)} \tag{17}$$

In Equations (16) and (17), $y_A^j$, $y_B^j$ and $y_{C_i}^j$ represent the $j$th element of vectors $Y_A$, $Y_B$ and $Y_{C_i}$, respectively, and $f_0$ is the mean of $Y_A$ elements (Equation (18)).

$$f_0 = \frac{1}{N}\sum_{j=1}^{N} y_A^j \tag{18}$$

Input space samples were generated using Latin Hypercube Sampling (LHS) [73] in the parameter ranges indicated in Table 2. Sample size was $10^6$ in all scenarios and the probability distribution was considered uniform for all inputs.In addition to computing main and total effects for each similarity index, bootstrapping was used to estimate confidence intervals for these sensitivity indices. 500 subsamples were used in all scenarios for bootstrapping. Finally, index convergence was assessed by sequentially increasing the number of model runs used to compute $S_i$ and $S_{Ti}$. The practical implementation of this method was conducted with help of the MATLAB toolbox provided by Pianosi et al. [74].

*3.4. Local Sensitivity Analysis*

In addition to GSA, local sensitivity analysis was conducted to gain further insight about metric performance around their nominal point of operation. Whereas GSA allowed the general assessment of metric behavior throughout the complete parameter input space, LSA facilitated a more detailed analysis of their application in practice. Requirements for image similarity measurement are not identical during and after registration of consecutive video frames. For the former, the metric must be robust and allow finding the point of maximum similarity. For the latter, the metric must provide a reliable absolute similarity estimation. Both applications were studied by means of local sensitivity analysis around the point of perfect alignment.

To achieve this, stable video sequences were sampled at an approximate frequency of 1 Hz, which was considered representative of typical operation conditions in a wildfire scenario. Each sampled frame was perturbed systematically through affine geometric transformations. Horizontal translations, vertical translations, rotations and scaling were applied separately. Their intensity varied sequentially within the ranges indicated in Table 2 in steps of 1% for translations, 1deg for rotations and 1% for scaling.

Once perturbed, each frame was compared to its original (i.e., stable) version to assess metric response to each movement component when the effect of all other factors—including scene variations and sampling frequency—was blocked. This approach was named *idealized operation conditions*.

Additionally, scene variations and time were considered by comparing each perturbed frame with the previous sampled frame. We refer to this approach as *realistic operation conditions*.

## 4. Results

This section summarizes and discusses the main results of this study. For the sake of readability, only a reduced subset with the most important results is included here. The interested reader can find a comprehensive compilation of all produced data for each considered scenario in Supplementary Materials.

Variance-based global sensitivity analysis techniques described in Section 3.3 were used to assess the general response of various image similarity metrics to six variables of interest, namely: horizontal translation, vertical translation, rotation, scale, time and sampling frequency. The first four parameters represent the camera movement to be detected, whereas time and frequency account for additional sources of image differences which may affect metric performance. An ideal image similarity measure should be highly sensitive to camera movement and robust in the presence of recording frequency variations and image content differences appearing over time.

### 4.1. GSA Convergence Considerations

Metric sensitivity was assessed using Main Effect (ME) and Total Effect (TE) indices as defined in Equations (14) and (15), respectively. ME and TE were estimated through Equations (16) and (17), with $N = 10^6$ LHS samples of the input space. This sample size was enough to achieve index convergence in all studied cases (see convergence results in Supplementary Materials).

However, ME and TE approximations converged to meaningless values in some cases (see Figure 4 for an example). This occurred when output similarity values did not follow a standard normal distribution, because $y$ values used in Equations (16) and (17) are expected to follow a standard normal distribution. Although output probability was approximately normally distributed in all cases, each similarity metric uses a different optimum value for perfect image match. As a result, output distributions provided by some similarity metrics were displaced with respect to the standard normal distribution.

This limitation was solved by centering the $y$ distributions provided by similarity metrics. Centering was achieved using Equation (19),

$$y_{centred} = (y - \overline{y})/\sigma_y \tag{19}$$

where $\overline{y}$ represents the average of $y$ and $\sigma_y$, its standard deviation. The application of Equations (16) and (17) to the centered distributions allowed the correct estimation of ME and TE. Figure 4 demonstrates the effect of centering $y$ distributions in a sample case. Verifying that model outputs approximately follow a standard normal distribution is essential to ensure correct estimation of variance-based sensitivity indices. However, common GSA libraries, including the code provided by Pianosi et al. [74], do not usually include the centering step by default. We therefore suggest paying special attention to this aspect and double check model output distributions obtained through Multi-Carlo sampling before proceeding further.

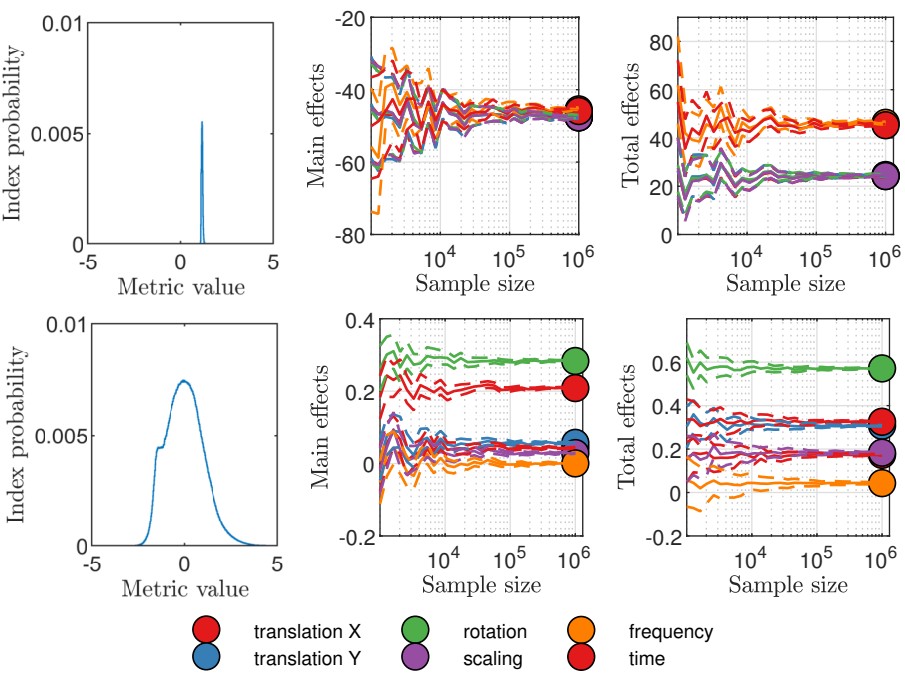

**Figure 4.** Effect of centering model output distributions computed during GSA. The top row shows sensitivity indices and their convergence obtained with the original distribution. The bottom row shows results achieved after centering the model output distribution. Mean values (solid lines) and confidence bounds (dashed lines) were estimated using bootstrapping with 500 resamples. Example results for Studholme's Mutual Information (SMI) in Scenario 1.

### 4.2. GSA Results

Average GSA results—obtained after solving issues with $y$ distributions—are displayed in Figure 5. According to them, all metrics had a similar response to frequency, whereas the strongest variation with time corresponds to IMSD and MI. On the contrary, 2D correlation showed the strongest response to all movement components while being less affected by time than IMSD and MI. NMI showed a better performance than MI, closely following 2D correlation.

One important conclusion that can be drawn from Figure 5 is the existence of important interactions among the six studied variables. While main effects account for the variance increase observed in model outputs when a single input parameter is varied, total effects include the effect produced by one parameter when the other input parameters are also allowed to vary. The fact that TE are significantly higher than ME demonstrates that individual parameter contributions to output variance are boosted when several input variables vary simultaneously.

The phenomenon of coupled response is especially important for 2D correlation, which could otherwise have been awarded the first place in this comparative study. High sensitivity to camera movement is ideal only if the source of image dissimilarity can be identified correctly. Conversely, there is little use in knowing that two images are different if the cause of this difference cannot be attributed to a single parameter. This fact reinforced the need for a local sensitivity analysis to gain further insight into model response to each individual parameter around the nominal point of operation.

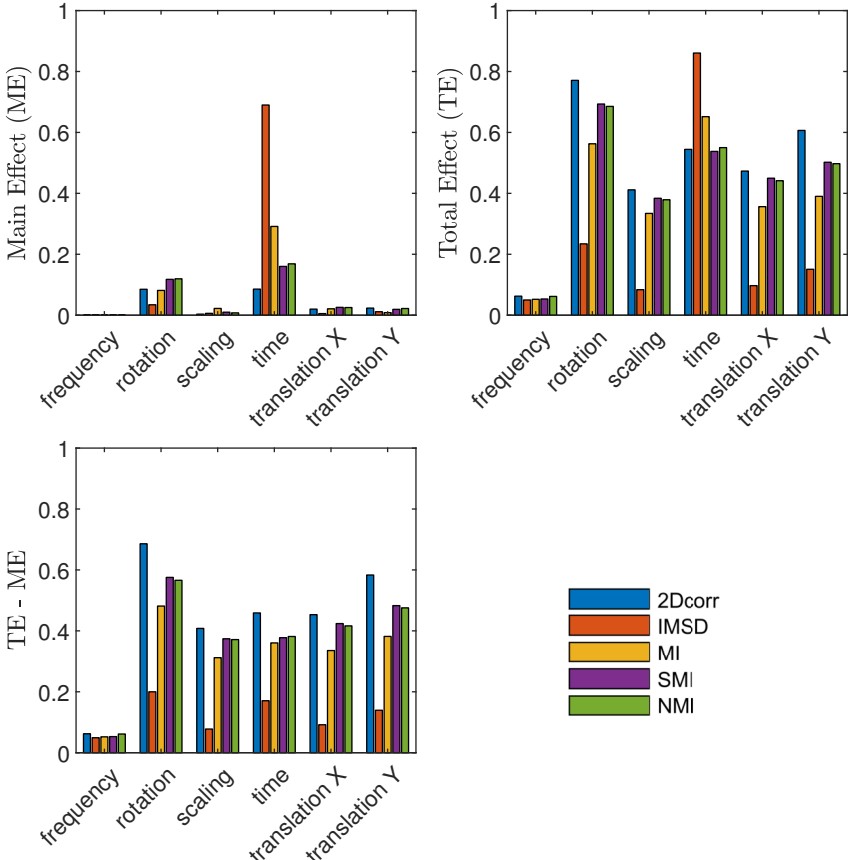

**Figure 5.** Global sensitivity indices of image similarity metrics to the six considered parameters, averaged over scenarios 1, 2, 4 and 5. Individual results for each scenario can be consulted in Supplementary Materials.

### 4.3. LSA Results

LSA was first used to assess existing differences in behavior between MI, SMI and NMI. Although their absolute values vary, all three metrics behaved similarly with camera movement (Figure 6). On the contrary, there was an important difference in their response to image content. Figure 7 displays metric behavior under idealized conditions in 3D. This representation highlights a significant variation of MI values with time. Considering the nature of fire TIR images, these differences are presumably due to the fact that the majority of the image entropy is provided by the fire. Consequently, image entropy increases as the image portion filled with fire grows over time. In its original formulation, Mutual Information between two images increases with the individual entropy of each of these images (see Equations (6)–(8)). Conversely, entropy normalization introduced in SMI and NMI cancels this effect (see Equations (9) and (11)).

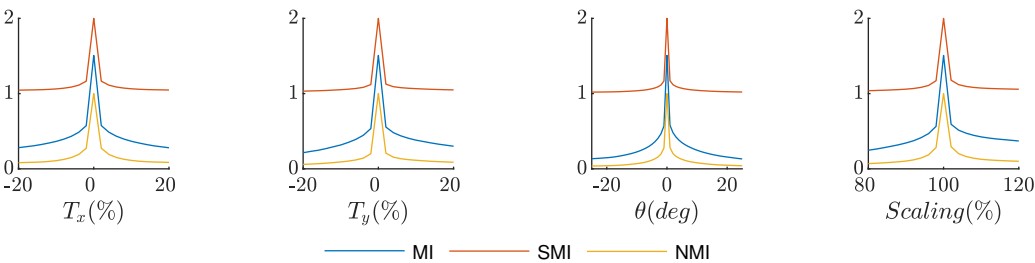

**Figure 6.** Local response of MI-based metrics to independent camera movement components under idealized conditions, i.e., when each video frame is compared to the stable version of itself. Averaged values along all studied video sequences. Camera movement components are: translation in the X direction ($T_x$), translation in the Y direction ($T_y$), rotation ($\theta$) and scaling.

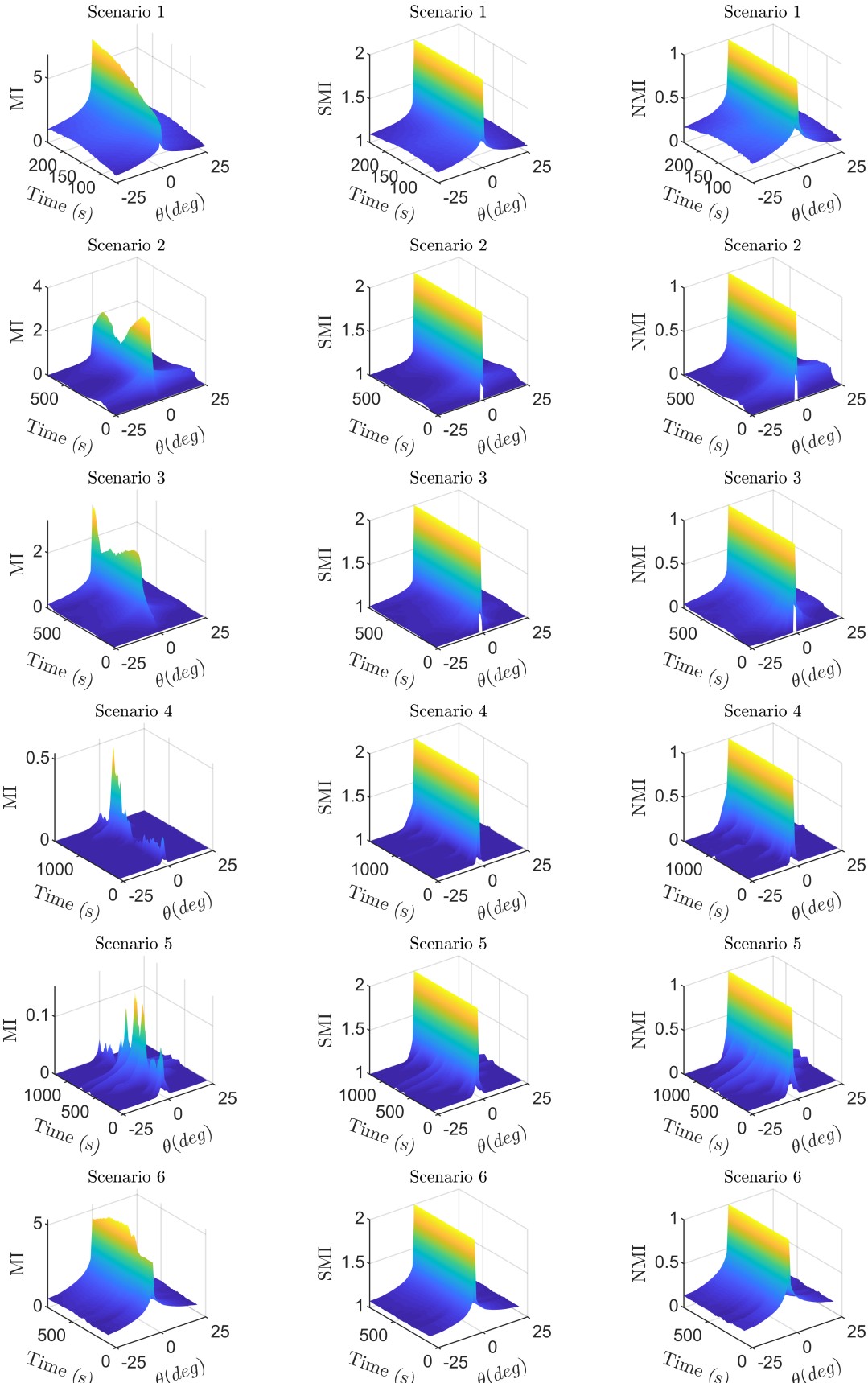

**Figure 7.** 3D representation of MI-based metrics response to time and image rotation under idealized conditions. Analogous results for translations and scaling are included in Supplementary Materials.

A further difference to be noted in Figure 7 is the maximum value achieved by each metric. Whereas SMI proved insensitive to time, Figure 7 demonstrates that it is indeed not normalized as its maximum value is not equal to one. NMI, while having a similar behavior, takes values in the restricted range $[0, 1]$, where 1 designates a perfect image match. Based on these results, NMI was deemed the best MI alternative for fire thermal image similarity analysis.

After selecting normalized mutual information among MI-based candidates, its performance was compared to 2D correlation and intensity mean squared difference. Figure 8 compares the average response of these metrics to synthetic camera movement. These results corroborate that IMSD is significantly less sensitive to image misalignments than NMI and 2Dcorr, as previously concluded from the global sensitivity analysis. Conversely, both NMI and 2Dcorr present an important peak at the position of perfect alignment, which makes them useful for image registration algorithms.

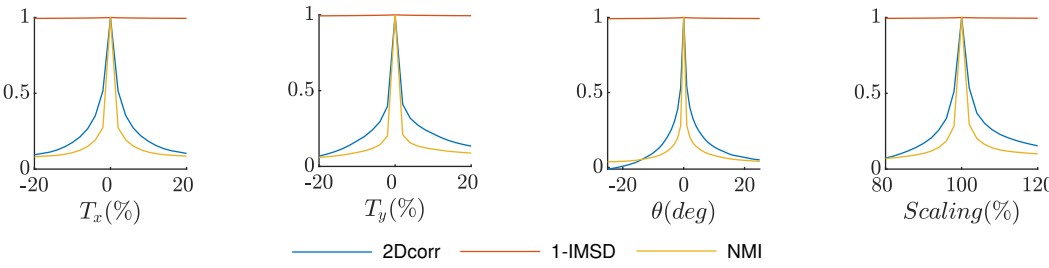

**Figure 8.** Metric response to independent camera movement components under idealized conditions, i.e., when each video frame is compared to the stable version of itself. Averaged values along all studied video sequences. Camera movement components are: translation in X direction ($T_x$), translation in Y direction ($T_y$), rotation ($\theta$) and scaling. 1-IMSD is displayed for consistency with the rest of metrics.

Nevertheless, results displayed in Figure 8 were obtained under idealized conditions in which each image was compared with a displaced version of itself. In a real scenario, the two IR images to be registered will not be identical. Typically, they will have been acquired from different perspectives or using dissimilar cameras. In a video stabilization problem, each video frame is to be compared to a previous frame of the same sequence. The amount of time elapsed between the acquisition of both frames will prevent an exact match even in the position of perfect alignment. This limitation affects similarity metrics differently, as demonstrated in Figure 9. Whereas 2D correlation can maintain optimum values close to 1 under real working conditions, NMI optimum values drop significantly due to the fact that a perfect image match is impossible. This behavior becomes more accentuated as sampling frequency diminishes, as shown in Figure 10.

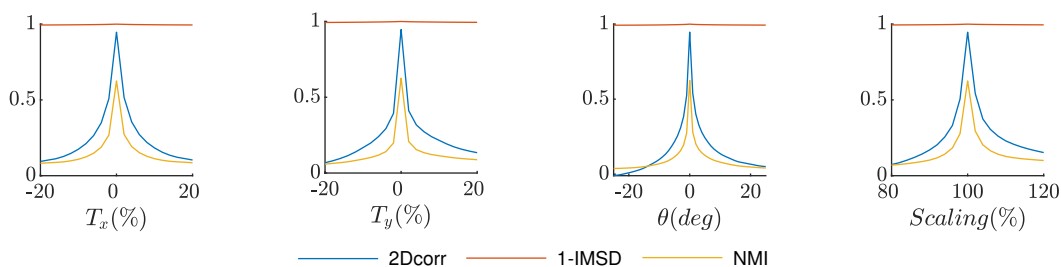

**Figure 9.** Metric response to independent camera movement components under realistic operation conditions, i.e., when each video frame is compared to the stable version of the previous frame. Averaged values along all studied video sequences. Camera movement components are: translation in the X direction ($T_x$), translation in the Y direction ($T_y$), rotation ($\theta$) and scaling. 1-IMSD is displayed for consistency with the rest of metrics.

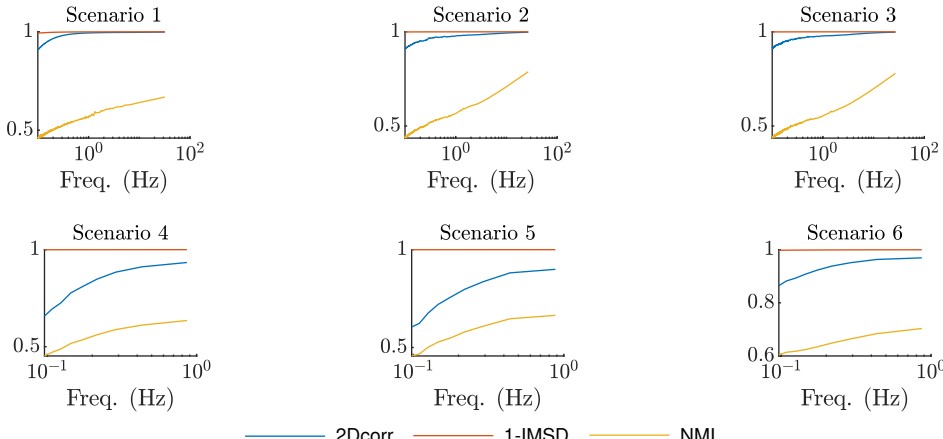

**Figure 10.** Metric response to video recording frequency. Displayed values show similarity between two stable consecutive frames, time-averaged along each video sequence. 1-IMSD is displayed for consistency with the rest of metrics.

Results displayed in Figure 9 highlight the first important difference in performance between 2D correlation and NMI. These results suggest that whereas both metrics can search for the perfect alignment position through an optimization strategy, they are not equally capable of assessing the quality of the achieved registration. In the specific case of IR fire video stabilization, 2D correlation values provide a reliable estimation of the alignment between consecutive frames. Two correctly registered frames will have a 2D correlation coefficient close to 1, whereas lower values can be attributed to misalignment. On the contrary, NMI may reach different maximum values depending on the video recording frequency, which prevents an absolute quality estimation. Therefore, we recommend the use of 2D correlation as a quality control metric *after* image registration.

To select the best-behaving metric *during* registration, one more property was analyzed: confidence of similarity values provided under real working conditions. Metric confidence was assessed first through their standard deviation when only one movement component was varied at a time (Figure 11). Additionally, metric robustness was analyzed in a general case in which the camera was allowed to move freely through a combination of translations, rotation and scaling (Figure 12).

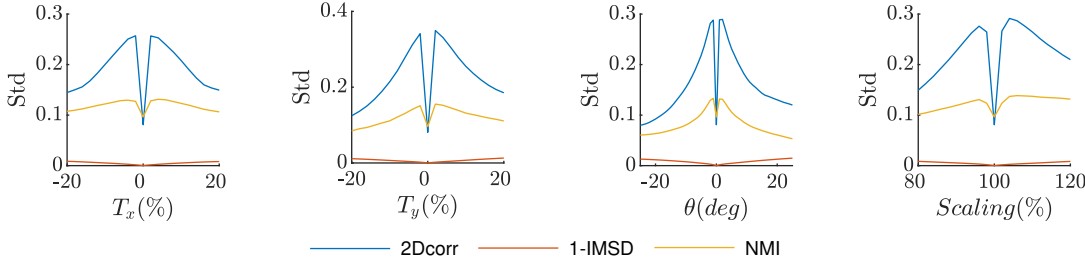

**Figure 11.** Metric value dispersion under real operation conditions, i.e., when each video frame is compared to the previous frame. Output standard deviation, computed along all studied video sequences, provides a quantitative assessment of how robust each metric is in front of natural image dissimilarities and recording conditions.

Figure 11 shows significantly higher standard deviation for 2D correlation than NMI, the only exception occurring at the point of perfect alignment. According to this, 2Dcorr values computed for a certain misalignment are subject to greater differences due to changes in image content and recording conditions. Therefore, confidence on image similarity estimation provided by 2D correlation can be considered lower in general. Interestingly, this does not hold for a small region around the point of perfect alignment, where 2Dcorr values became more precise. Based on these results, 2Dcorr

can still be considered suitable for robust estimation of achieved registration quality, whereas NMI outperformed 2Dcorr in situations far from perfect alignment, i.e., during registration.

These results are supported by Figure 12, which shows the statistical difference in similarity values computed under real and idealized conditions. Such difference was assessed through Bland-Altman plots, a common tool widely used to compare results provided by two methods designed to measure the same property. Bland-Altman plots are built by graphically displaying measurement differences along the complete range of measured values. Bias and limits of agreement are superimposed to the scatter plot. Bias is computed as the average difference, whereas limits of agreement are estimated as bias plus and minus 1.96 times the difference standard deviation [75]. Finally, both bias and limits of agreement are accompanied by their respective 95% confidence intervals, which were computed here using the approximated estimations proposed by Bland and Altman [76]. Confidence intervals are not always computed in the literature when using Bland-Altman plots, although they have been considered essential by some authors [77].

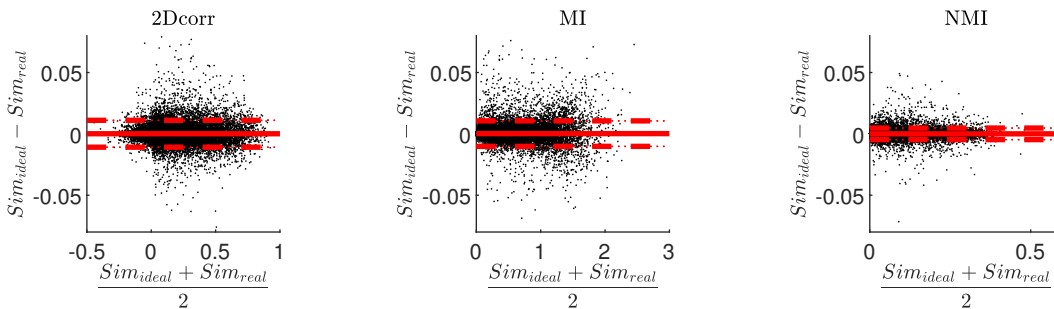

**Figure 12.** Bland-Altman plots comparing metric behavior under real and idealized conditions. $Sim_{ideal}$: similarity measured between each perturbed frame and the stable version of itself; $Sim_{real}$: similarity measured between each perturbed frame and the previous stable frame. Black dots are individual random samples along all studied scenarios; red solid lines indicate mean bias; red dashed lines indicate Limits of Agreement (LoA); red dotted lines represent 95% confidence intervals for estimated bias and LoA. Wide LoA are representative of significant sensitivity to changes in the reference frame used for registration.

Figure 12 supports the hypothesis that NMI is more robust than 2D correlation under real conditions, especially when various movement components are combined. Although no significant bias was appreciated in any method, narrower limits of agreement mean that similarity estimations provided by NMI under real conditions are in general closely related to estimations provided under idealized conditions. This implies that NMI sensitiveness to changes in the reference frame (Figure 9) is limited to a small region around the point of perfect alignment. On average, when considering the complete camera movement space, NMI similarity estimations were more robust than those provided by 2D correlation and MI.

For this reason, we propose the use of NMI as the best performing image similarity metric for inter-frame registration in video stabilization frameworks. Maximization of normalized mutual information is likely to cancel misalignments for a wide range of video recording frequencies. Therefore, we encourage the application of optimization algorithms such as the ones proposed by Chen et al. [42] or Kern and Pattichis [39] for MI. However, optimum NMI achieved close to the point of perfect alignment cannot be used as a reliable estimation of absolute registration quality. Instead, we recommend using the 2D correlation coefficient for absolute alignment measurement.

## 5. Discussion

Our results are aligned with previous findings published in related fields. Due to its higher robustness to outliers and noise, Mutual Information has been selected as the primary similarity metric for image registration in multiple applications including medical imaging [46,78], stereo processing [79] and object

tracking [80,81], among others. Previous comparative studies also found that MI produces consistently sharper optimum peaks at the correct registration values than correlation [82]. We observed a similar behavior in fire TIR imagery.

The use of the 2D correlation coefficient as a quality control metric is not so common in previous literature. When new algorithms are developed, registration accuracy is usually measured in a controlled environment where ground-truth camera movement is known or synthetically generated. In these cases, algorithm performance is assessed by comparing predicted and ground-truth registration transformations [39,44,83–86]. However, this approach cannot be used for quality control in a general operational scenario where the ground-truth registration transformation is unknown.

Other authors used pixel gray value mean absolute difference or root mean squared difference to assess registration quality [87]. However, similarity metrics based on gray differences are highly sensitive not only to the image relative position but also to the image content, as demonstrated here. Given two pairs of images with the same relative position, gray difference metrics are likely to give a higher registration rating to the image pair with lower contrast. We therefore recommend the use of 2D correlation for this purpose instead.

Given these results, the most immediate follow-up work for this paper consists in using the selected methods to build an IR video stabilization system suitable for aerial wildfire monitoring. In addition, this study could be extended to other image similarity metrics not considered here. We assessed and discussed some of the most popular methods in their basic formulation. However, there exist a wealth of variants that were derived from the basic algorithms to improve their performance. Among other adaptations, several authors have proposed the use of multiresolution schemes to measure similarity in image registration problems [80,88]. These methodologies provide additional improvements and their behavior should be analyzed.

## 6. Conclusions

This article analyzed alternative approaches to measure image similarity within a TIR fire image registration framework. Image registration is an important pre-processing step for the study of wildfire behavior through remote sensing. Within registration, image similarity measurement requires special attention because the estimation of image misalignment is essential to accomplish image registration as well as to control its quality.

Performance of any image similarity metric is highly dependent on the specific application for which it is used. For this reason, the primary goal of this study was the assessment of general similarity measurement approaches for the specific problem of fire thermal image registration. Image registration requires image similarity measurement for two different purposes. First, image similarity is treated as a cost function during the optimization problem linked to registration. Secondly, a robust estimation of absolute image similarity is essential for quality control. The highly dynamic nature of any wildfire scenario adds important difficulties when comparing images acquired at different times. This can result in mismatches at any time, even with the most accurate registration method. Such outliers must be automatically detected if the algorithm is meant to work unsupervisedly.

Without an in-depth analysis, it may seem that the methodology used to estimate image similarity does not have further implications as long as metrics provide higher similarity values for better aligned images. Results presented here show that this is not the case and careful attention should be paid to the method used to measure quality of alignment. Our results demonstrate that different image similarity metrics are affected differently by camera translation, rotation, distance to fire, size of fire, recording frequency and temporal changes in the scene. Such distinct behavior may motivate the selection of one metric or another depending on the specific goal to be achieved. In the case of video stabilization, we suggest using Normalized Mutual Information (NMI) for similarity maximization during inter-frame registration, whereas the 2D correlation coefficient is recommended for absolute alignment assessment and quality control.

These results constitute a key departing point for further studies into remote sensing of active wildfires through aerial TIR imagery. Furthermore, we described a generic and systematic methodology that can be replicated for analogous studies. GSA and LSA are usually complementary, and so were they in this study. While LSA provided detailed insight into metric behavior around the nominal operation point, GSA allowed measurement of the general sensitivity of metric candidates to factors that can vary widely and do not have fixed nominal values, such as scene content and video sampling frequency.

Finally, our analysis of MI-based metrics may help in other image registration problems. NMI inherits MI strengths and solves MI limitations regarding sensitiveness to image overlap and dependence on absolute image entropy. Because MI-based metrics allow general comparison between multi-modal images without making assumptions about their nature, NMI may be a powerful metric for multi-modal image registration or image fusion. As with thermal cameras, multi- and hyper-spectral sensors are becoming more compact and lighter, which will boost their potential for near-range remote sensing in wildfire operations. In a likely future scenario, several remote sensing platforms may be flying simultaneously over an active fire, each one carrying different sensors and acquiring complementary views at different times from different perspectives. How to fuse these data will undoubtedly be a topic of interest in the near future, and studies such as the one we present here contribute towards a better understanding of image processing alternatives.

**Supplementary Materials:** The following are available online at http://www.mdpi.com/2072-4292/12/3/540/s1, Figures S1–S4: GSA index convergence in each scenario, Figure S5: Global sensitivity indices in each scenario, Figures S6–S10: LSA of MI-based metrics under idealized conditions in each scenario, Figure S11: LSA results under idealized conditions in each scenario, Figure S12: LSA results under real conditions in each scenario, Figure S13: Metric value dispersion under real conditions in each scenario, Figure S14: Bland–Altman plots comparing metric behavior under real and idealized conditions in each scenario.

**Author Contributions:** Conceptualization, M.M.V. and S.V.; methodology, M.M.V. and S.V.; software, M.M.V.; validation, C.M., E.P. (Elsa Pastor), E.P. (Eulàlia Planas), M.M.V., O.R. and S.V.; resources, D.J., E.P. (Elsa Pastor), E.P. (Eulàlia Planas), L.Q. and S.V.; data curation, C.M., D.J., L.Q. and M.M.V.; writing—original draft preparation, M.M.V.; writing—review and editing, all authors; visualization, C.M., M.M.V. and O.R.; supervision, E.P. (Elsa Pastor), E.P. (Eulàlia Planas) and S.V.; project administration, E.P. (Elsa Pastor) and E.P. (Eulàlia Planas); funding acquisition, M.M.V., E.P. (Elsa Pastor) and E.P. (Eulàlia Planas). All authors have read and agreed to the published version of the manuscript.

**Funding:** This research was funded by the Spanish Ministry of Education, Culture and Sport (Grant FPU13/05876), the Spanish Ministry of Economy and Competitiveness (projects CTM2014- 57448-R and CTQ2017-85990-R, co-financed with FEDER funds), the Erasmus+ Traineeship Program and Obra Social La Caixa research mobility grants.

**Acknowledgments:** The authors thank Valentijn Hoff for his help with data collection and Bret Butler for his support during this research and his comments on the manuscript draft.

**Conflicts of Interest:** The authors declare no conflict of interest. The funders had no role in the design of the study; in the collection, analyses, or interpretation of data; in the writing of the manuscript, or in the decision to publish the results.

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
