# Peer review of "Image Similarity Metrics Suitable for Infrared Video Stabilization during Active Wildfire Monitoring: A Comparative Analysis"

_remotesensing, doi:10.3390/rs12030540_

Round 1

Reviewer 1 Report

This is an interesting research about video image frame registration using thermal video imagery. The manuscript needs to be revised according to the comments/suggestions made below before considering it for publishing.

The authors need to provide some figures to:

Better explain the domains of the six experimental fire scenarios studied. Show some examples of misalignment between image frames and corresponding rectified image frames.

Author Response

A new figure has been added with the experimental setups (figure 3).

Regarding misalignment and registration examples, it must be noted that we did not perform image registration in this study. The six considered video sequences were originally stable and there was no misalignment between consecutive frames. Although we did apply synthetic movement to the original frames, we did so in a systematic (and repetitive) manner. There is already a figure in the manuscript (fig. 2) with sample frames from all video sequences. Perturbed frames are, to the naked eye, very similar to the ones shown in figure 2 and an extra figure would add very little information here. For this reason, and because there are already 12 figures in the paper, we would prefer not add another one.

Reviewer 2 Report

This paper is a study of different similarity metrics that can be used for registration of fire infrared images.  The descriptions of image similarity metrics and sensitivity analysis approaches are clear and thorough, and the experiments and results are well-designed and well-presented.  Although no novel approach is presented, the insights gained from comparing the different metrics and the comprehensive background material provide a valuable contribution to the application of wildfire monitoring.

Author Response

We are grateful for this assessment of our work. As no other recommendation was made, we took the opportunity to revise the English language and style, also suggested by other reviewers.

Reviewer 3 Report

The authors compare image similarity metrics for sequence of images obtained using thermal infrared images over six experimental fires. The topic is of interest and is applicable for remote sensing wild fires.  The article is well written and the introduction is clean and comprehensive.  The authors test a few known methods for registering TIR images with fires and draw conclusions from their test scenarios and recommend procedures.

I feel I am not qualified to comment on their methodology, however I have two major concerns on their approach.

TIR is not the only wavelength available for use with a drone or aerial Imagery.  If there are NIR ~ 0.8um/ SWIR (1-2um) sensors, the fires are reduced to only active flaming fronts instead of the significantly larger areas covered using TIR wavelengths (for example check out the publications pasted below).  So instead of trying to register single band images, a more practical method would be to use other spectral bands that are already co registered, and that are less sensitive to fire. The manuscript completely ignores this and builds on registering thermal images only.

Further, is unclear if the image similarity metrics is a basis of registering images with fires as there is no illustrative example.  The article as such is very lengthy and perhaps most can go as supplemental information, while an illustrative example of image registration is lacking in the manuscript.

Schroeder, W., Prins, E., Giglio, L., Csiszar, I., Schmidt, C., Morisette, J. and Morton, D., 2008. Validation of GOES and MODIS active fire detection products using ASTER and ETM+ data. Remote Sensing of Environment112(5), pp.2711-2726.

Kumar, S.S., Picotte, J.J. and Peterson, B., 2019. Prototype Downscaling Algorithm for MODIS Satellite 1 km Daytime Active Fire Detections. Fire2(2), p.29.

Author Response

Although it is true that cameras with other spectral bands could provide better images of the active fire front, the reality is that the most used ones, because they are more affordable, are the ones working on the TIR band. Moreover, if only one camera is to be installed in the drone, the TIR band allows to obtain information on other aspects of the fire, related to the smouldering and burned zone. Most of the existing IR aerial images of real and experimental fires are also on the TIR band. So, even though working in other bands is also interesting, to have methods that provide image similarity metrics for TIR images is definetively very important in the field of wildfires.

Reviewer 4 Report

The paper is very interersting work and is well written but the level of english language must be improve and the references must be update to more actual research study.

Author Response

We have conducted a thorough revision of the English language and style. We have also included the following recent references related to the studied similarity metrics in section 2:

Kaneko et al, 2002 Yang et al., 2020 Wu et al, 2019 Li et al., 2020 Eikhosravi et al., 2020 Liu et al., 2019 Jones et al., 2019 Lüdemann et al., 2019 Xu et al., 2008 Panin et al, 2012 Zhuang et al., 2016

We are not aware of any more recent work about fire monitoring applications or sensitivity analysis techniques that we had not cited already.

Reviewer 5 Report

The title of the manuscript (MS) deals with Image Similarity Metrics suitable for Infrared VideoStabilization During Active Wildfire Monitoring: a Comparative Analysis. Generally, I found this to be an interesting and generally well-written manuscript. I have a number of relatively easily addressed comments detailed below.

Specific Comments:

The Abstract should be re-written/re-organized. Currently, it reads like part of the introduction.
"Usually, it is one paragraph of 300 words or less, the major aspects of the entire paper in a prescribed sequence that includes: 1) the overall purpose of the study and the research problem(s) you investigated; 2) the basic design of the study; 3) major findings or trends found as a result of your analysis; and, 4) a brief summary of your interpretations and conclusions."

In the introduction,
Line 43: "In these cases, the remote sensor is placed in a fixed position or a hovering aircraft and it is deployed to collect high spatial resolution images with moderate temporal resolution for the full duration on flaming combustion" needs a reference, also add "a" before "moderate"...

In the "Result and Discussions" section the authors must extend the comparison between their approach and other ones that have been developed and used in the literature for the same or related purposes (I recommend increasing the number of Scientific articles cited, especially to compare the study context with similar studies). Also, in this section, the authors should also highlight the current limitations of their study, and briefly mention some precise directions that they intend to follow in their future research work.

Author Response

Response to the Specific Comments:

The Abstract should be re-written/re-organized. Currently, it reads like part of the introduction.
"Usually, it is one paragraph of 300 words or less, the major aspects of the entire paper in a prescribed sequence that includes: 1) the overall purpose of the study and the research problem(s) you investigated; 2) the basic design of the study; 3) major findings or trends found as a result of your analysis; and, 4) a brief summary of your interpretations and conclusions."

Great observation. The abstract has been re-written following these guidelines.

In the introduction,
Line 43: "In these cases, the remote sensor is placed in a fixed position or a hovering aircraft and it is deployed to collect high spatial resolution images with moderate temporal resolution for the full duration on flaming combustion" needs a reference, also add "a" before "moderate"...

Three references have been added to support this statement and the edit has been made.

In the "Result and Discussions" section the authors must extend the comparison between their approach and other ones that have been developed and used in the literature for the same or related purposes (I recommend increasing the number of Scientific articles cited, especially to compare the study context with similar studies). Also, in this section, the authors should also highlight the current limitations of their study, and briefly mention some precise directions that they intend to follow in their future research work.

An additional “Discussion” section has been added to accommodate this analysis. We included 13 more references of related studies and compared their findings with ours in this section. The main limitations of our study as well as suggested future work have also been highlighted in the Discussion section.

Round 2

Reviewer 3 Report

" Although it is true that cameras with other spectral bands could provide better images of the active fire front, the reality is that the most used ones, because they are more affordable, are the ones working on the TIR band. Moreover, if only one camera is to be installed in the drone, the TIR band allows to obtain information on other aspects of the fire, related to the smouldering and burned zone. Most of the existing IR aerial images of real and experimental fires are also on the TIR band. So, even though working in other bands is also interesting, to have methods that provide image similarity metrics for TIR images is definetively very important in the field of wildfires."

 Most drones almost invariably have an RGB camera  in addition to the IR/TIR/ MIR.   I agree, payload weights are serious considerations and the aim is always to reduce weight, however they also have necessary sensors that help  getting their location and orientation which are also good methods to register images. 

While i am not underscroing the work by authors for registering single band images , I am pointing out that that a  discussion on other existing and practical options in their manuscript will be nice.

I recommend that the authors include a short paragraph discussing these and substantiate it with relevant citations. 

Reviewer 5 Report

Thank you for the thorough consideration of my comments, and the excellent additions to the manuscript.